# In-Situ Thermography Investigation of Crack Growth in Armco Iron under Gigacycle Fatigue Loading

Victor Postel [1,2], Johann Petit [2], Chong Wang [1,3,*], Kai Tan [3], Isabelle Ranc-Darbord [2], Qingyuan Wang [1,*] and Daniele Wagner [2]

1.  Failure Mechanics and Engineering Disaster Prevention and Mitigation Key Laboratory of Sichuan Province, Sichuan University, Chengdu 610207, China; victor.postel@hotmail.com
2.  Laboratoire Energétique Mécanique Electromagnétisme, UPL, Université Paris Nanterre, 50 Rue de Sèvres, 92410 Ville d'Avray, France; johann.petit@parisnanterre.fr (J.P.); isabelle.ranc@parisnanterre.fr (I.R.-D.); daniele.wagner@parisnanterre.fr (D.W.)
3.  Department of Mechanics, Sichuan University, Chengdu 610207, China; tkiu1289244086@163.com
*   Correspondence: chongwang@scu.edu.cn (C.W.); wangqy@scu.edu.cn (Q.W.)

**Abstract:** A non-destructive thermographic methodology based on the temperature field is utilized to determine the crack tip position during the very high cycle fatigue (VHCF) test of pure iron and deduce the corresponding fatigue crack growth rate (FCGR). To this end, a piezoelectric fatigue machine is employed to test 1 mm thick pure iron samples at 20 kHz in push–pull fatigue loading. Two cameras are placed on each side of the plate sample, an infrared one for measuring the temperature fields on the specimen surface and an optical one for visualizing the crack tip verification. The centre section of the specimen is notched to initiate the crack. The temperature field is converted into intrinsic dissipation to quantify the inelastic strain energy according to energy conservation. The maximum value of intrinsic dissipation in each thermal image is related to the position of the crack tip and thus allows monitoring of the crack evolution during the fatigue test. The obtained results show that one specific specimen broke at $7.25 \times 10^7$ cycles in the presence of a very low-stress amplitude (122 MPa). It is observed that the intrinsic dissipation has a low-constant level during the initiation and the short cracking, then sharply grows during the long cracking. This transition is visible on the polished surface of the sample, where the plasticity appears during the long cracking and slightly before. The material parameters in the Paris equation obtained from the intrinsic dissipation in the short crack growth are close to the results available in the literature as well as those obtained by the optical camera.

**Keywords:** very high cycle fatigue; ultrasonic fatigue test; intrinsic dissipation; fatigue crack growth rate; thermography

## 1. Introduction

There is a growing need to extend the service life of systems and components well beyond the traditional fatigue design limit of $10^7$ cycles, commonly known as the very high cycle fatigue (VHCF) regime. Owing to the accelerating frequency up to 20 kHz with an ultrasonic fatigue test machine [1], many VHCF experiments have been performed in a reasonable time. Even under very small stress amplitudes [2], many deformations can occur within the material microstructure and initiate cracking [3,4]. The initiation of cracks in VHCF for metallic materials without inclusions occurs during sliding along crystallographic planes, characterized by the presence of irreversible slip bands or persistent slip bands on the material surface [5,6]. The crack initiation can also take place in grain boundaries and form flat surfaces—the so-called grain-boundary cracking [7]. Furthermore, crack propagation goes through several distinct stages. In the first stage, there is the initiation, which is closely associated with the material microstructure [7]. The second stage is the short cracking, where the plasticity level slightly develops, and the crack slowly

grows [8]. The last stage is the long cracking, characterized by a steadily increased growth speed, plasticity visible on the sample's surface, and striations in the fracture surface [9]. The fatigue life in the VHCF regime is mainly consumed by the crack initiation, which corresponds to more than 99% of the lifespan [10,11]. Thereby, a significant portion of the fatigue life involves micro-scale mechanisms in response to cyclic stresses [3,12], which means that the steady growth just consumes a few of the entire life. Multistage propagation and minor fatigue life in steady crack propagation are the two distinguishing features of the VHCF. Nevertheless, a brief survey of the literature reveals the lack of knowledge on the direct evidence and physical mechanism of the short crack in the VHCF.

Many techniques exist to assess the fatigue crack growth rate (FCGR), which is essential for predicting the lifetime of cyclically loaded materials to process a damage tolerance design [13,14]. The position of the crack tip in macroscale structures is usually evaluated by the crack tip opening displacement (CTOD) [15], optically by analysing the outer surface of the sample [16,17], by analysing the striations on the fracture surface [18], or acoustic emission [19], or by an energetic approach, such as the energy dissipation [20] or total cyclic plastic dissipation [21]. In situ technologies, such as micro-X-ray [22], transmission electron microscope (TEM) [23], or synchrotron [24], allow the scrutiny of crack propagation in a higher resolution without disrupting the tests. The technologies cited were mainly conducted in the low cycle fatigue regime rather than the VHCF because of the difficulty of the long-term acquisition method during ultrasonic fatigue testing. To calculate the FCGR in the VHCF, He et al. [25] measured the crack length replication by pausing the tests. Another approach is to analyse the striation pattern on the fracture surface after the final fracture occurrence [26]. Ouarabi et al. [13] obtained the parameters of the Paris equation in the VHCF of three steel sheets by examining the crack propagation based on optical images of the external surface of the samples. However, little experimental work has been conducted on the crack growth rate in the VHCF regime. Furthermore, other than crack length detection, developing an appropriate method to recognize damage status and the crack stage is of great importance for structural integrity monitoring.

During the service life of a component, the material suffers from many damages, commonly resulting in stress concentration at defects, such as notches. Due to the small deformation magnitude during the fatigue test, inspecting the crack tip without a hypersensitized instrument would be difficult. When a cracked specimen experiences cyclic loading mode, a cyclic plastic zone appears in front of the crack tip [21,27]. The deformations produced in the plastic zone in front of the crack yield a temperature rising [28] that can be more easily identified by a thermal camera. Thermographic methods allow to characterize the behaviour of materials [29] subjected to different loads (for example cyclic loads). Thermal effects in fatigue can estimate the fatigue life of materials [30]. The development of the energetic method facilitates the use of infrared thermography in predicting the fatigue behaviour of materials under uniaxial and multiaxial loading with constant amplitude [31–33]. The studies carried out by Bär et al. [34] demonstrated the relationship between the plastic zone and the energy dissipated at the crack tip during fatigue tests. According to Meneghetti et al. [35], the maximum temperature can be measured in the centre of the plastic zone, ahead of the crack tip. The energy dissipated in heat at the crack tip allows following the evolution of the crack during tensile [36] and fatigue tests [21]. An indicator called intrinsic dissipation is calculated to evaluate the irreversible deformation. It has been examined in zero-dimensional [37], one-dimensional [38,39], two-dimensional (2D) [32,40], and three-dimensional domains [41]. The fatigue crack velocity from the dissipation values was elaborated on 316L steel at the low cycle fatigue regime by Wang et al. [18]. This method employs the maximum measured value to define the crack tip, allowing the crack length and FCGR to be measured. Therefore, intrinsic dissipation provides an applicable in situ scheme to examine the crack stages from the thermodynamic viewpoint.

In the present study, FCGR tests are conducted on an Armco iron specimen. The position of the crack tip for in situ specimens is detected by an optical camera and by the pixel's position at the maximum level of intrinsic dissipation. The elaboration of the Paris

law enables us to correlate the results obtained by the two methods. Patterns obtained from multistage fractographic are associated with different levels of energy dissipation.

## 2. Experimental Procedures

A piezoelectric fatigue machine with in situ observation equipment was employed to achieve the VHCF domain at the appropriate time cost (see Figure 1). The ultrasound generator produces a wave that propagates through the sample to vibrate it. The tests were carried out at a frequency of 20 kHz with a stress ratio $R = \sigma_{min}/\sigma_{max} = -1$. The specimen has thickness of 1 mm. It vibrates together with its longitudinal harmonic model in tension–compression. Two cameras were placed in the middle of the sample, where the stress was maximum, and the displacement was null, allowing to take clear surface images with a reasonable magnification. The geometry of the sample is illustrated in Figure 2. The surface was electropolished to avoid artificial defects and to accommodate optical monitoring. The average roughness *Ra*, measured by a nanoindentation test (KLA-iNano, KLA, Milpitas, CA, USA) near the external surface of the specimen near the notch, is 24.519 nm. As shown in Figure 2, the specimen has a spherical notch in its centre to cause crack initiation at a specific location. A fatigue crack is generally nucleated from the notch due to the stress concentration introduced by the geometric discontinuity [42].

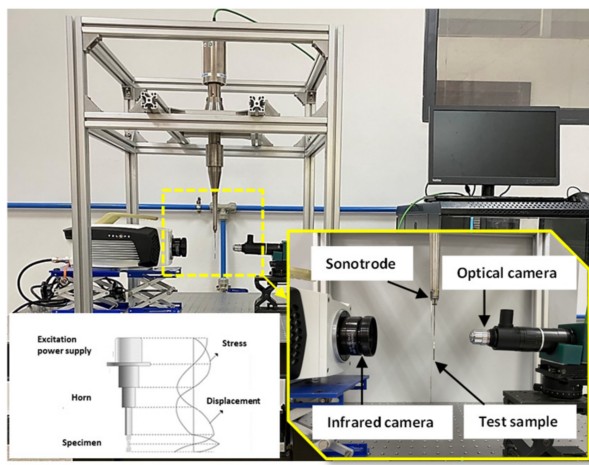

**Figure 1.** In situ ultrasonic fatigue test machine.

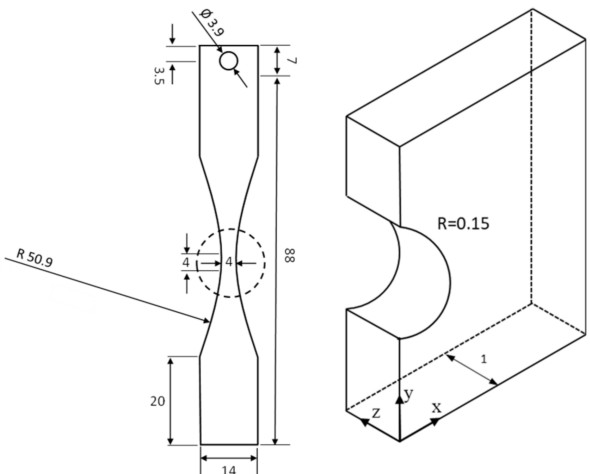

**Figure 2.** Dimensions (in mm) and design of the specimen.

During post-processing, the crack tip position was measured on every frame from two cameras to calculate the crack length and then used to calculate the crack growth rate.

An optical camera (ace acA200-50gm, Basler, Ahrensburg, Germany) and An InfraRed camera (Fast M200, Telops, Quebec City, QC, Canada) were employed for observation. The spatial resolution of the infrared camera was 14 μm per pixel, and the acquisition frame rate was 100 fps. The optical camera was about 0.4 μm (×280) per pixel, with an acquisition frequency of 80 Hz. The purpose of this examination is to focus on the initiation and the short crack propagation. The magnitude of the applied stress was 122 MPa based on previously conducted tests by Wang [43], expecting the number of cycles greater than $1 \times 10^7$ cycles. The tests were performed at room temperature with pressurized air blown on the test tube to escape excessive self-heating.

The studied material is a polycrystalline Armco iron from Goodfellow (Cambridge, England), whose chemical composition is given in Table 1. It is a 99.8% pure iron, known as a typical BCC metal, widely utilized to scrutinize the mechanisms of crack initiation and propagation [44–46]. Due to the low carbon content, the microstructure consists of 100% ferrite. An EBSD analysis done before the experiment showed an average grain size of 30 μm without particular orientation. The material was tested in its receiving condition. The modulus of elasticity and ultimate tensile stress in order were 207 GPa and 290 MPa, as provided in the Supplementary Materials.

**Table 1.** Chemical composition of Armco iron (%).

| C | Mn | P | S | Cu | N | Si | Al | Cr | Mo | Ni | Sn |
|---|---|---|---|---|---|---|---|---|---|---|---|
| 0.001 | 0.050 | 0.005 | 0.003 | 0.007 | 0.0022 | 0.003 | 0.005 | 0.014 | 0.002 | 0.013 | 0.002 |

## 3. Theory and Formulation

Due to any discontinuity in the material (notch, crack, or any defect), stress concentration leads to inhomogeneous deformation while undertaking a force. During the fatigue test, the inelastic deformation inside the material generates heat. During plastic deformation, the heat source is known as energy dissipation. The relationship between fracture mechanics and thermodynamics during fatigue tests has been studied by many scientists [47–50]. It is manifested by a variation in temperature when the material is subjected to external solicitations. As presented in Figure 3, the externally exerted stress $p$ results in deformation at the crack tip, leading to a temperature difference.

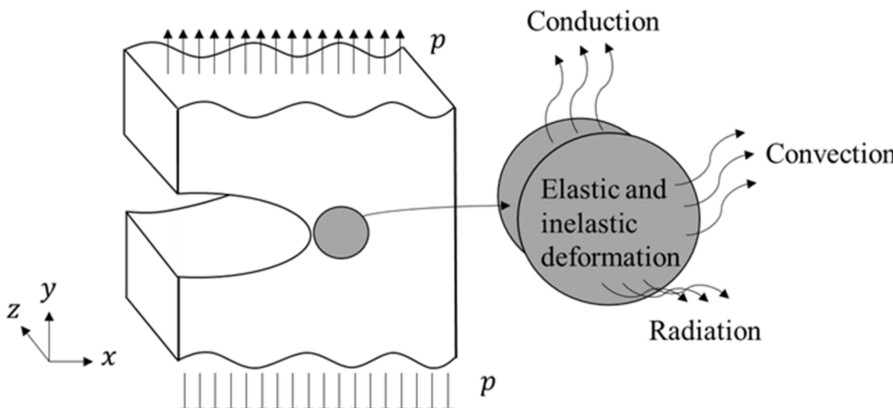

**Figure 3.** Thermomechanical exchanges at the crack tip. The externally applied force is converted into elastic and inelastic (plastic) deformation, in which the resulted heat is exchanged with the environment under convection, radiation, and conduction.

During fatigue loading, accounting for the hypothesis of small deformation, local plasticity ahead of the crack tip appeared, leading to a temperature variation. The temperature at any point of the sample can be affected by the internal thermal conduction and the thermal exchanges with the ambient environment [51]. Based on the properties of the material and the temperature field, another indicator, known as intrinsic dissipation



($d_1$), was also evaluated and considered in the analysis. Intrinsic dissipation is defined as the energy dissipated (W·m$^{-3}$) due to the irreversible microstructure deformation [52]. Combining the local expressions of the first and second principles of thermodynamics and using Fourier's law, the local 2D heat equation can be written as [20,38,53,54]:

$$\rho C \left( \frac{\partial \bar{\theta}}{\partial t} + \frac{\bar{\theta}}{\tau^{2D}} \right) - k \left( \frac{\partial^2 \bar{\theta}}{\partial x^2} + \frac{\partial^2 \bar{\theta}}{\partial y^2} \right) = \sigma : \dot{\varepsilon}_p = d_1(t) + S_{the}(t) + r \tag{1}$$

where $\theta$, $d_1$, and $S_{the}$, are the temperature difference ($T$–$T_0$) on the material surface, the intrinsic dissipation, and the thermomechanical coupling, respectively. The time constant $\tau^{2D} = \rho C e / 2h$ represents the heat loss by convection and radiation between the specimen surfaces and the surrounding medium calculated on the whole specimen area and from all frames after the final failure. Equation (1) is calculated from the material characteristics $\rho$ (density), $C$ (specific heat capacity), the isotropic conduction tensor k, and the thickness of the specimen e. The thermomechanical coupling (i.e., $S_{the}$) is set equal to zero or neglected [55] because the heat sources related to thermoelasticity balance each other over a cycle during a fully reversed tension–compression fatigue test (R = −1) [56]. The external volume heat source r is considered null because no external heat source is applied. The term $\partial\theta/\partial t$ is the temporal derivative of the temperature. The term $\partial^2\theta/\partial x^2 + \partial^2\theta/\partial y^2$ represents the 2D Laplacian of the temperature field in Cartesian coordinates. Based on the balance of linear moment, the energy conservation, and the second law of thermodynamics, the intrinsic dissipation represents the relationship between the Cauchy stress tensor $\sigma$ and the plastic strain tensor $\varepsilon_p$. The analysis of intrinsic fatigue dissipation was performed by Blanche et al. [38] on copper to show that intrinsic dissipation exists whatever the attainable stress range. An energetic method provides a rapid evaluation of fatigue parameters based on the intrinsic dissipation [54] and the microplasticity of the material [55]. Intrinsic dissipation is also exploited to assess dissipative effects during fatigue testing [56–58].

## 4. Results and Discussion

Several experiments have been conducted and they behave similarly. In the present study, one specimen, that was broken at $7.25 \times 10^7$ cycles under the action of a very low-stress amplitude (122 MPa), is presented below as representative.

### 4.1. Temperature

Figure 4 shows the evolution of the maximum temperature measured on the specimen's surface as a function of the number of cycles. This curve has been widely investigated in previous studies [48] and consists of three apparent branches. During the beginning of the test (phase I), the temperature rapidly rises. Then, the material adapts to the applied stress, and the temperature growth is followed by a lower rate (phase II). During phase III, the temperature rises very quickly until arriving at the specimen's final failure. Our study focused on the number of cycles beyond $7 \times 10^7$, from which the crack starts to propagate. Two images were chosen to show the crack evolution from the thermography point of view (see Figure 4), corresponding to $7.193 \times 10^7$ and $7.208 \times 10^7$ cycles. These points are placed on the evolution graph of the maximum temperature measured in each image as a function of the number of cycles. The black arrows are toward the particular position of the pixel, with the highest temperature measured in each image.

### 4.2. Intrinsic Dissipation

The intrinsic dissipation ($d_1$) is calculated using Equation (1) and the frames of the temperature field. For this purpose, a numerical code was developed and allowed to obtain the intrinsic dissipation in the 2D spatial domain of the problem. A temperature image and the corresponding image of $d_1$ at the same cycle number (at $7.193 \times 10^7$ cycles) are shown in Figure 5a,b. In each image of $d_1$, the pixel with the maximum value corresponds to the position of the crack tip, so it is possible to search for this information over a wide area.

This procedure is a full-field analysis, which is convenient for observing a large area. The $d_1$ from the global values, and not only the surrounding of the maximum pixel, provides the localization of the crack tip, even with a low magnification of the IR camera. On the contrary, an optical camera requires a high magnification to increase the contrast to have sufficient sensitivity.

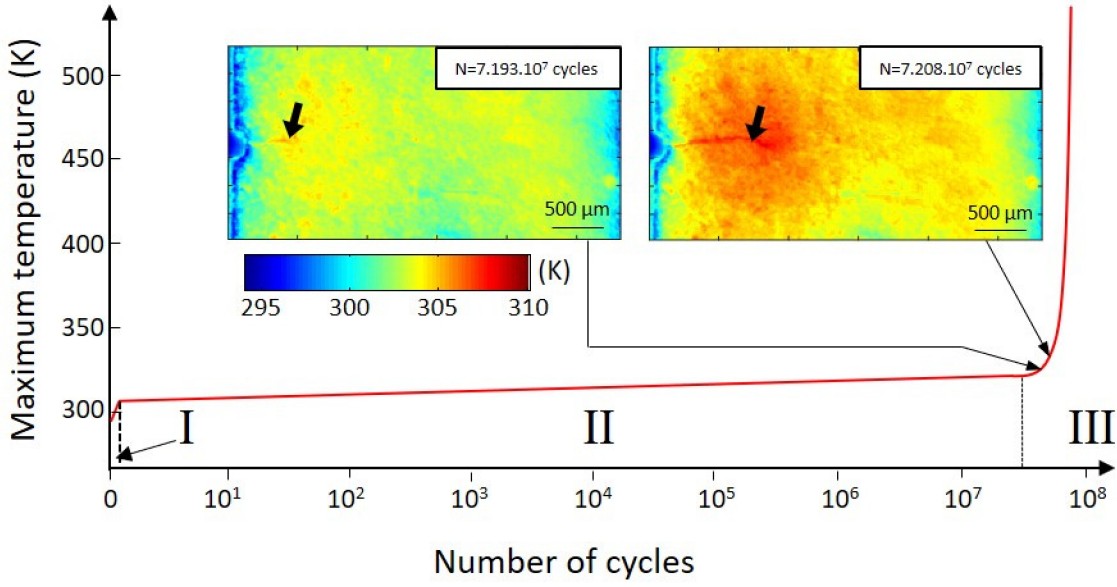

**Figure 4.** Distribution of temperature fields for various numbers of cycles (those pertinent to the crack tip are specified by the black arrows), and the evolution of the maximum temperature (°K) versus the number of cycles.

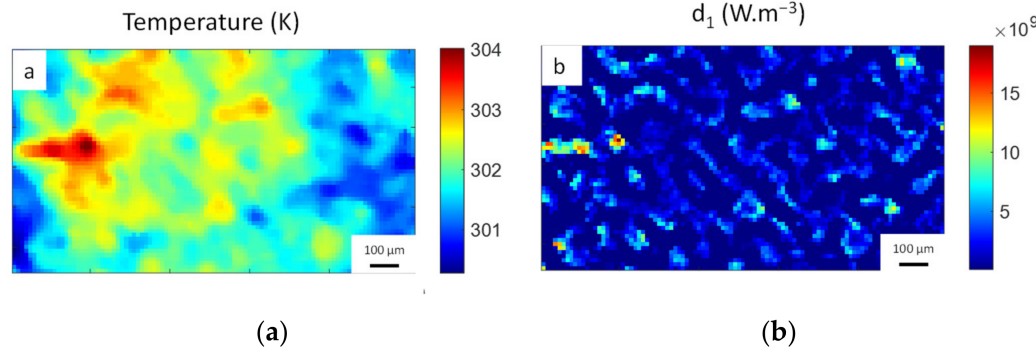

(a)                                                                  (b)

**Figure 5.** (**a**) Temperature field graph; (**b**) intrinsic dissipation graph at $7.193 \times 10^7$ cycles.

According to the thermodynamic relation in the heat equation given above, the temperature evolution at a location relates not only to deformation but also to temperature field, convection, conduction, and radiation. On the contrary, the intrinsic dissipation will remain steady if irreversible deformation keeps at the same level. Figure 6 plots the evolution of the maximum value on temperature (in black) and $d_1$ (in blue) as a function of the number of cycles from $7.180 \times 10^7$ to $7.205 \times 10^7$ cycles. It is found that the average of temperature would slightly increase, while the average of $d_1$ remains relatively stable. Indeed, from the thermodynamic point of view, it is in good agreement that the temperature tends to continuously increase in the case of unvarying d1 and the negative sign of the Laplacian (see Equation (1)). Then, both curves (temperature and intrinsic dissipation) sharply increase from $7.205 \times 10^7$ cycles.

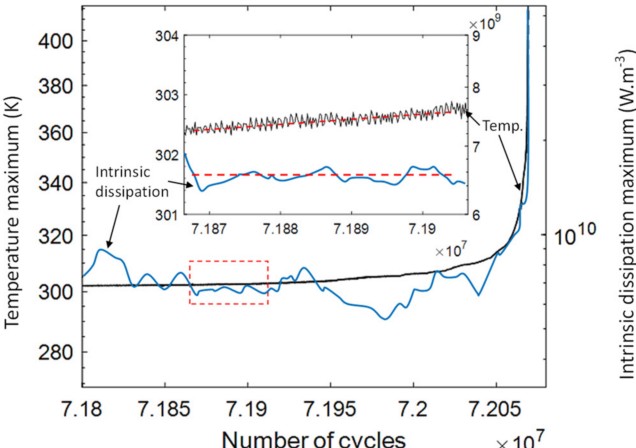

**Figure 6.** Evolution of the maximum temperature and the maximum intrinsic dissipation per image in terms of the number of cycles.

Wang's experiments [18] showed the relationship between the fatigue crack growth rate and the intrinsic dissipation of a 316 steel. The tests were performed on CT specimens at 10 Hz with three different loading ratios *R*, i.e., 0.1, 0.3, and 0.5. The order of magnitude of the intrinsic dissipation was around $10^7$ W·m$^{-3}$, which is lower than our results. Our calculated values for $d_1$ were around $10^{10}$ W·m$^{-3}$ for the stable area. Wang's tests were conducted at 10 Hz with an acquisition frequency of the IR camera at 80 Hz. Since the acquisition frequency during our tests was lower than that of the fatigue machine (100 Hz for the camera and 20 kHz for the machine), the recorded images included several cycles. Therefore, the calculated intrinsic dissipation corresponded to 250 cycles. Taking into account the number of cycles per image, we obtained the same level order of intrinsic dissipation as that reported by Wang et al. [18].

*4.3. Graphy*

The fatigue crack surface fractography obtained by scanning electron microscope (SEM) after the final fracture highlights the different mechanisms of crack propagation in the fatigue process: the initiation stage, and the short and long crack propagation (see Figure 7). The transitions are defined based on the SEM observation of the fracture surface. The arrow indicates the direction of crack propagation, and the direction is the same for all provided images in the subfigures of Figure 7. In the image given in Figure 7a, the red dotted curve delimits the transitional zone between the initiation and the short cracking (at 252 microns on the optical camera side and 181 microns on the IR camera side). The red curve corresponds to the transition between the short cracking and the long cracking (at around 1409 microns). The initiation phase is characterized by the grain trace appearance on the fracture surface [6,43] and flat surface (Figure 7b), whose origin is attributed to an intergranular cracking [7]. Following the initiation, there is a short cracking phase, characterized by the presence of ridges (see Figure 7c). This pattern appears because the crack is less sensitive to the material's microstructure [4]. After developing the crack for a specific length, depending on the stress amplitude and material, the crack changes from short to long crack propagation. The long cracking is generally characterized by striations [18,59] on the fracture surface, as demonstrated in Figure 7d, where ten striations are achieved at a length of 4.6 μm.

By observing the outer surface of the specimen after the final fracture occurred (Figure 8a), the enlarged pictures during the short cracking (Figure 8b,c), and at the long cracking (Figure 8d), it is possible to notice that the specimen presents little or no plasticity scenario during the short cracking, however, a massive number of slip bands appear during the long cracking (Figure 8d). The white line in Figure 8b represents a boundary between a deformed area and a non-deformed one. It appears as though the deformation did not enter into the adjacent grain. In comparison,

Figure 8c presents no evidence of deformation. A small amount of the visible plasticity zone in Figure 8b may explain the slight differences in $d_1$ in Figure 6.

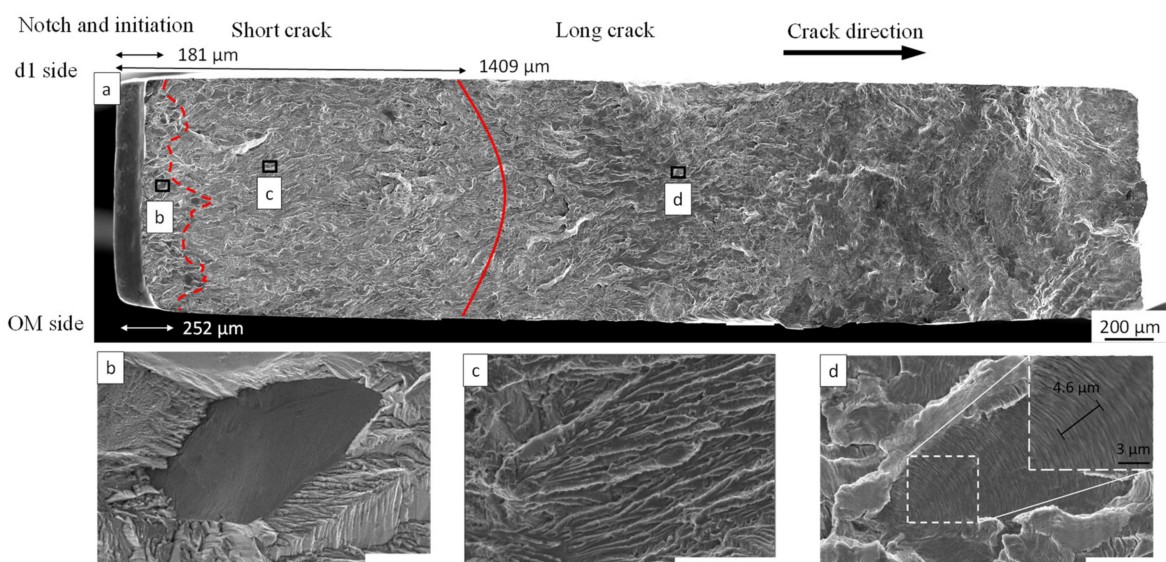

**Figure 7.** SEM observations on the fracture surface after final failure. (**a**) Full scale of fatigue crack surface. The red dashed and solid lines delimit the boundaries between the initiation and the short cracking, and the short cracking and the long cracking. (**b**) The intergranular cracking at the initiation stage. (**c**) The ridge patterns during the short crack stage. (**d**) The classic striations pattern in the stable propagation stage with a crack growth of about 0.46 μm/cycle.

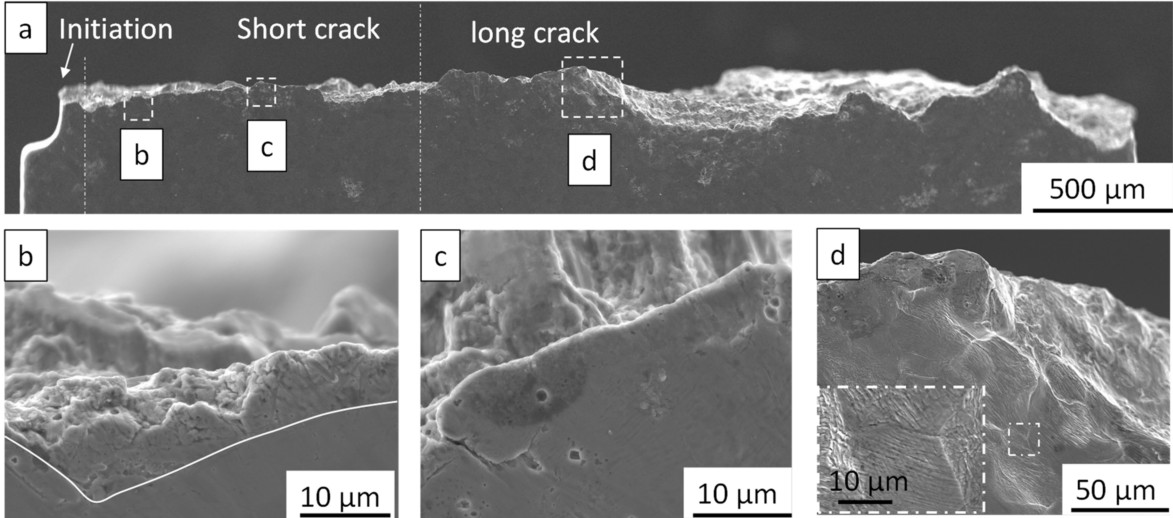

**Figure 8.** Wide view of the outer surface of the specimen after the occurrence of the final fracture (**a**) and three enlargements in the short cracking (**b**,**c**) show little plasticity, and the long cracking (**d**) presents much plasticity.

### 4.4. Evolution of the Crack Tip Position

An optical camera recorded the specimen surface during the fatigue test to assess the crack tip position. The crack length is measured along the horizontal and vertical axes; however, the present scrutiny focuses on the values along the x-axis. The field covered by the camera is 700 μm wide to focalize the study on the initiation and short crack propagation. As reported previously in Section 4.2, an infrared camera measures the temperature and then the $d_1$ parameter. Figure 9 shows the crack length evolution

obtained from both approaches in terms of the number of cycles. The blue and red dotted lines show the limit between the two stages of propagation (initiation and short cracking) on each side of the specimen. The transitions between the initiation and short cracking, specified by the red and blue dotted lines in Figure 9, are determined from the fractographic results in Figure 7. After $7.170 \times 10^7$ cycles, the crack similarly propagated on each side of the specimen. Although the results are very similar, the propagation on each side of the specimen is not perfectly synchronized, which causes differences in the crack lengths.

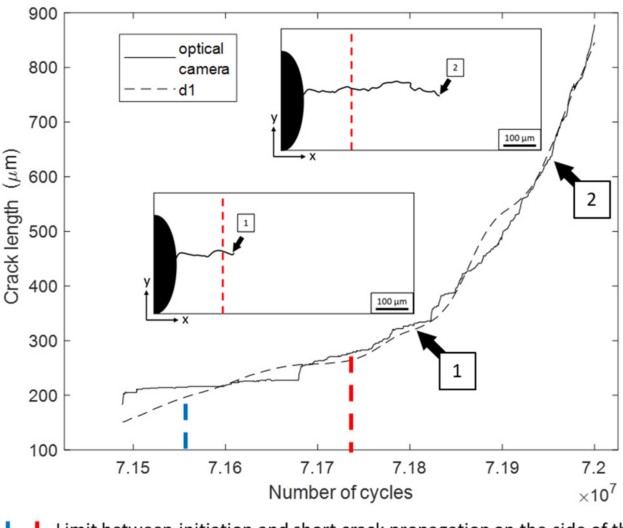

Limit between initiation and short crack propagation on the side of the specimen with the optical camera (red) and the side of the d1 (blue)

**Figure 9.** Evolution of the crack length in terms of the number of cycles. Two instant of crack growth that marked point 1 and 2 illustrate evolution of crack length and path.

The evolution of the crack length measured from the position of the maximum $d_1$ in each 2D image and the level of $d_1$ in terms of the number of cycles is demonstrated in Figure 10. The crack length remains increasing while the $d_1$ shows two distinct areas. The first is with stabilized and reasonably low values of $d_1$ (specified by the blue colour), and the others (presented by the red and orange colours) are a rapid increase of $d_1$. Two corresponding lengths of fatigue crack could identify at each transition from one to another. It shows that the transition on intrinsic dissipation is from crack length 1398 µm to 1731 µm. The first value is very close to 1409 µm, founded by the fractographic observations (see Figure 7a), and marks the passage from the short cracking phase to the long cracking.

The generated heat sources are related to the production of plastic deformation [60]. Hence, a low level of $d_1$ indicates a low amount of plasticity on the specimen's surface, in agreement with the low level of plasticity visible in Figure 8c. While the level of d1 is higher for the red and orange zone, indicating plastic deformation in front of the crack tip. For the number of cycles greater than $7.208 \times 10^7$ cycles, the magnitude of d1 increases continuously. From this length, the plastic deformation can appear on several grains, and the more the crack progresses, the more the deformed area increases, as shown in the two images of the fracture surface in Figure 10. After 1731 µm, widespread plastic deformation was identified on the surface of the specimen. The red zone in Figure 10 indicates that the transition from the short crack to the long crack takes place over a certain distance, and in this zone, the plasticity level increases, as does the $d_1$. This critical investigation revealed that the level of $d_1$ can be exploited as an important indicator to disclose the state of the crack propagation. On the contrary, the evolution of the crack length in Figure 10 increases continuously with the number of cycles. It is fairly impossible to distinguish the above-mentioned three areas with only information on the crack length, which is unconcerned by the pattern features on fractography and the crack opening mechanisms.

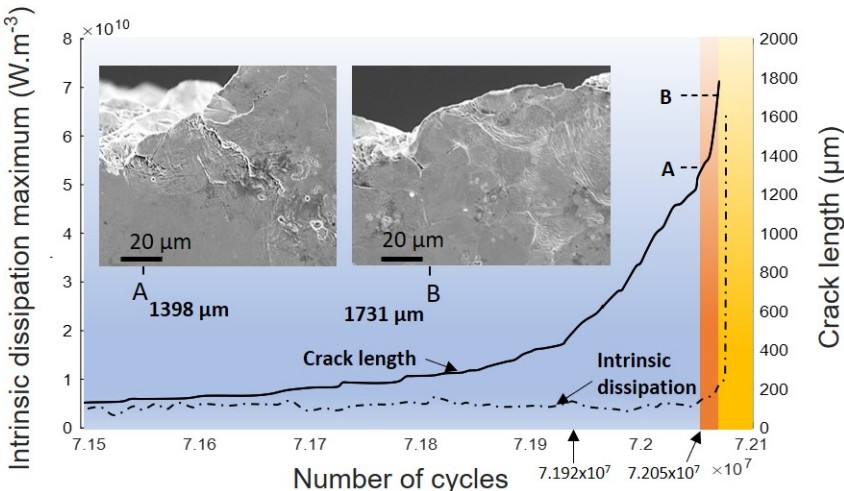

**Figure 10.** Evolution of the crack length based on the position of the maximum of intrinsic dissipation $d_1$ versus the number of cycles, and evolution of the level of $d_1$ versus the number of cycles.

### 4.5. Discussion

4.5.1. Comparison between the Optical and Thermography

The stress intensity factor range ($\Delta K$) for an edge crack within a semi-infinite body under the action of cycling loading at $R = -1$ is calculated from the following formula [17,61,62]:

$$\Delta K = K_{max} - K_{min} = f\left(\frac{a+b}{W}\right)\Delta\sigma\sqrt{\pi(a+b)} \tag{2}$$

$$f\left(\frac{a+b}{W}\right) = 1.12 - 0.231\left(\frac{a+b}{W}\right) + 10.55\left(\frac{a+b}{W}\right)^2 - 21.7\left(\frac{a+b}{W}\right)^3 + 30.382\left(\frac{a+b}{W}\right)^4 \tag{3}$$

where the factor $a$ is the crack length, $b$ is the depth (165.35 µm on the optical microscope side, and 150.2 µm on the $d_1$ side) of the notch, $W$ denotes the width of the specimen, and $\Delta\sigma$ represents the stress range. Figure 11 presents the fatigue crack growth rate (m/cycle) results versus the stress intensity factor range $\Delta K$ (MPa·m$^{1/2}$) based on the crack lengths displayed in Figure 9.

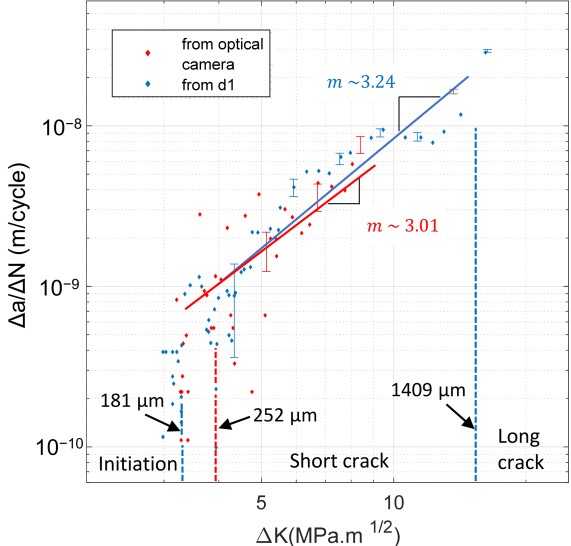

**Figure 11.** The evolution of the fatigue crack growth rate (m/cycle) in terms of the stress intensity factor ranges from the optical results (red dots) to $d_1$ (blue dots).

Figure 11 shows the fitting of the experimental data of the FCGR in terms of the stress intensity factor based on the optical camera measurements and the $d_1$ approach in a double logarithm coordinate system. The dotted lines correspond to the transition from the fractographic images in Figure 7a. According to Figure 7a, the transitions (initiation to short cracking) appear at 252 microns for the optical camera and 181 for the infrared camera. Furthermore, based on Equations (2) and (3), the $\Delta K$ of the transition lengths are at 3.92 MPa·m$^{1/2}$ for the $d_1$ results and 3.29 MPa·m$^{1/2}$ for the optical camera results. Whereas the fatigue tests performed by Pippan et al. [60] on Armco iron revealed the existence of a threshold value of $\Delta K$ at 2.75 MPa·m$^{1/2}$. The transition between the short and the long cracking (at 1409 μm) is equivalent to a $\Delta K$ of 15.0 MPa·m$^{1/2}$. These results allowed us to establish the dotted lines, as presented in Figure 11. A trend line calculation was performed on the part associated with the short cracking. The results, expressed in the power form, are characterized by the well-known Paris equation, $\Delta a/\Delta N = C \cdot \Delta K^m$, where the parameters $C$ and $m$ are the material constants [63]. The results for each method are obtained as follows:

From $d_1$: $\Delta a/\Delta N = 1 \times 10^{-11} \, \Delta K^{3.24}$

From the optical camera: $\Delta a/\Delta N = 2 \times 10^{-11} \, \Delta K^{3.01}$

The value of the exponent $m$ in Figure 11 is 3.24 (in blue) for the optical method and 3.01 for the $d_1$-based methodology (in red). Tests performed by Lieurade et al. [64] on C–Mn steels in the air at low-frequency show FCGR results as follows in Table 2:

**Table 2.** The values of C and m parameters are based on the Paris law on C–Mn steels [64].

| C (m·cycle$^{-1}$·MPa·m$^{1/2}$) | m |
|---|---|
| $3.9 \times 10^{-12} < C < 5.32 \times 10^{-11}$ | $2.53 < m < 3.15$ |

Wang's tests [21] on 316L steel show that the values of m range from 3.36 to 3.55. The results were obtained from the thermographic assessment of the specimen surface by employing the maximum value of energy dissipation at the crack tip position. Tests were carried out by Ouarabi et al. [13] under conditions similar (tension–compression, cooled, $R = -1$, $f = 20$ kHz) to those of our tests but conducted on CP1000 steel. The obtained results show a Paris equation obtained from the optical analysis of the crack as follows: $\Delta a/\Delta N = 3 \times 10^{-12} \, \Delta K^{3.27}$. The results of the Paris equation measured in ultrasonic fatigue from the optical or $d_1$ methods are in fairly good agreement with those found in the literature.

4.5.2. Uncertainty

Since one of the crucial objectives of the present work is to rationally measure the crack growth rate of Armco iron, it is relevant to quantify the experimental uncertainties. The crack propagation speed is calculated from the following formula: $V = \Delta a/\Delta N$. Therefore, the uncertainty in the measurement of the speed ($u_{\Delta a/\Delta N}$) is calculated as follows:

$$u_{\Delta a/\Delta N} = \sqrt{\left(\frac{1}{\Delta N}\right)^2 u_a^2 + \left(\frac{\Delta a}{(\Delta N)^2}\right)^2 u_{N,pro}^2} \tag{4}$$

$$u_{N,prop} = \sqrt{f_{ave}^2 \times u_T^2 + T^2 \times u_{f,ave}^2} \tag{5}$$

where $T$ is the time in seconds, the frequency resolution of the infrared camera is 1/100 of a second, therefore, the time uncertainty corresponds to $u_T$ would be 0.01 s. The average value of $f$, $f_{ave}$, represents the average frequency of the fatigue crack propagation during the loading block and is equal to 20 kHz. The uncertainty on the frequency $u_{f,ave}$ is equal to the standard deviation around the mean frequency, where $u_{f,ave} = 1$ Hz. The uncertainty in the measurement of the crack tip position $u_a$ on the infrared camera corresponds to the

cumulative uncertainties of 1 pixel between the position of the crack in an initial image and a final image, calculated by: $u_a = \sqrt{2} \times 14 \times 10^{-6} = 1.98 \times 10^{-5}$ m.

The uncertainty values are presented in Figure 11, where the evolution of the FCGR as a function of the stress intensity factor is plotted. As the crack length increases, the relative uncertainty decreases. According to Table 3, after the appearance of a crack with a small length, the measurement uncertainty becomes small enough to guarantee the accuracy of the measured crack growth rate values.

**Table 3.** Uncertainty on the FCGR for different crack lengths based on the obtained results from $d_1$.

| Crack Length (μm) | $\Delta K_{eff}$ (MPa·m$^{1/2}$) | $\Delta a/\Delta N$ (m/cycle) | $u_{(\Delta a/\Delta N)}$ (m/cycle) | Uncertainty Relative (%) |
|---|---|---|---|---|
| 300 | 4.3359 | $8.6945 \times 10^{-10}$ | $2.9298 \times 10^{-8}$ | 58.5 |
| 500 | 5.9186 | $4.1389 \times 10^{-9}$ | $5.1187 \times 10^{-10}$ | 12.3 |
| 700 | 7.5353 | $6.2420 \times 10^{-9}$ | $5.1565 \times 10^{-10}$ | 8.2 |
| 900 | 9.3140 | $9.0510 \times 10^{-9}$ | $5.2131 \times 10^{-10}$ | 5.7 |
| 1100 | 11.3245 | $8.5580 \times 10^{-8}$ | $5.2708 \times 10^{-10}$ | 6.1 |
| 1300 | 13.6817 | $1.6358 \times 10^{-8}$ | $5.3729 \times 10^{-10}$ | 3.3 |
| 1500 | 16.3508 | $2.9298 \times 10^{-8}$ | $5.4037 \times 10^{-10}$ | 1.8 |

## 5. Conclusions

A new approach is proposed for determining crack propagation based on the intrinsic dissipation ($d_1$) evolution at the onset of fatigue loading in mode I. Two methods have been employed to measure the crack tip position in fatigue tests with very high numbers of cycles. The results focus on the initiation and the short cracking, which mainly corresponds to the beginning of the crack propagation. From the results presented earlier in this article, it can be concluded for the test condition that:

1.  The conversion temperature into the intrinsic dissipation allows us to realize the damage state during crack propagation. During the short cracking, very little plasticity is visible at the material's surface, and the level of $d_1$ is low. On the contrary, plasticity is predominant during the long cracking, and the $d_1$ increases sharply. This issue indicates that the $d_1$ value can be implemented as an indicator of the crack length.
2.  The obtained results show that there exists a physical correlation between intrinsic dissipation and deformations. This fact is more apparent at the material surface than those obtained based on the thermal field. The temperature level keeps increasing during the short cracking, while the intrinsic dissipation shows a stable level, while the plasticity level does not grow during the short cracking.
3.  The small size of a crack makes it challenging to be detected by an optical camera even with high magnification. During the beginning of the crack propagation, the generated heat source at the crack tip facilitates its detection by an infrared camera. By utilizing full-field analysis, the maximum value of $d_1$ on 2D images of the sample surface is a suitable method to indicate the position of a crack, thus measuring its length and defining its FCGR.

**Supplementary Materials:** The following supporting information can be downloaded at: https://www.mdpi.com/article/10.3390/met12050870/s1, Figure S1: Fracture surface of sample Armco 8 to 11 with indication of three propagation stages. Figure S2: EBSD result together with crack path and growth rate of sample Armco 8. Figure S3: EBSD result together with crack path and growth rate of sample Armco 9. Figure S4: EBSD result together with crack path and growth rate of sample Armco 10. Figure S5: EBSD result together with crack path and growth rate of sample Armco 11. Figure S6: EBSD result together with crack path and growth rate of sample Armco 12. Figure S7: The crack growth rate versus the stress intensity factor for the 5 experiments. Figure S8: Comparing of crack growth rate by the trend curves on 5 samples. Table S1: Threshold of cyclic SIF at transition from short to long crack for the 5 specimens.

**Author Contributions:** Investigation, V.P.; validation, J.P.; funding acquisition and conceptualization, C.W.; data curation, K.T.; methodology, I.R.-D.; supervision, Q.W.; administration and review, D.W. All authors have read and agreed to the published version of the manuscript.

**Funding:** National Natural Science Foundation of China (No. 12022208, No. 11832007, No. 11972021) and the Sichuan Science and Technology Program (No. 2019YJ0158).

**Institutional Review Board Statement:** Not applicable.

**Informed Consent Statement:** Not applicable.

**Data Availability Statement:** The study did not report any data.

**Acknowledgments:** The authors acknowledge Yan Li for providing helpful guides on the performed experiments.

**Conflicts of Interest:** The authors declare no conflict of interest.

## Nomenclature

| | |
|---|---|
| $\rho$ | Density |
| $C$ | Specific heat capacity |
| $r$ | External heat source |
| $k$ | Isotropic conduction tensor |
| $h$ | Heat transfer coefficient |
| $e$ | Thickness of the specimen |
| $\bar{\theta}$ | Temperature difference ($T-T_0$) on the material surface |
| $T_0$ | Temperature reference |
| $d_1$ | Intrinsic dissipation |
| $\tau^{2D}$ | Heat loss by convection and radiation |
| $S_{the}$ | Thermomechanical coupling |
| $\sigma$ | Cauchy stress tensor |
| $\varepsilon_p$ | Plastic strain tensor |
| $u_{\Delta a/\Delta N}$ | Uncertainty on the measurement of the crack growth rate |
| $u_{N,prop}$ | Uncertainty on the number of cycles |
| $f_{ave}$ | Average value of the frequency |
| $\Delta N$ | Cycle number variation |
| $\Delta K$ | Stress intensity factor range |
| $\Delta a$ | Crack length variation |
| $u_T$ | Time uncertainty |
| $u_a$ | Uncertainty in the measurement of the crack tip position |
| $T$ | Time |
| $u_{f,ave}$ | Uncertainty on the measurement of the frequency |
| $a$ | Crack length |
| $b$ | Depth of the notch |
| $W$ | Width of the specimen |
| $C$ | Material constant |
| $m$ | Material constant |

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
