# Peer review of "In-Situ Thermography Investigation of Crack Growth in Armco Iron under Gigacycle Fatigue Loading"

_metals, doi:10.3390/met12050870_

Round 1

Reviewer 1 Report

The title is very interesting and the manuscript written well. Also, it has a good structure. The results are very valuable and finally, I recommended to publish the manuscript as the present form. But, the authors should pay attention to some points:

1-It is strongly suggested to provide figures 1 and 2 in more resolution.

2- related to the preparation of fatigue specimens, it is stated that specimens were electropolished. It is better to report the surface roughness of the samples. Because the roughness is an important factor affect fatigue properties of metallic materials.

3-on page 4, lines 131-134, the microstructure, initial grain size, and tensile properties of raw material are presented. So, please refer to an appropriate reference or describe about the obtaining ways of them. 

Author Response

1-It is strongly suggested to provide figures 1 and 2 in more resolution.

 Answer: Many thanks to the suggestion on resolution. The two figures have been remade from images with higher resolution and saved in an uncompressed format. The figures are presented in the revised manuscript and response file.

2- related to the preparation of fatigue specimens, it is stated that specimens were electropolished. It is better to report the surface roughness of the samples. Because the roughness is an important factor affect fatigue properties of metallic materials.

Answer: The roughness of the sample was measured by Nano indentation testing in a 50 by 50 microns area near the notch. The average roughness value is 24.519 nm (see figure below). An explanatory sentence has been added in the text of the publication:

The surface was electropolished to avoid artificial defects and to accommodate optical monitoring. The average roughness Ra , measured by a nano indentation test (KLA- iNano) near on the external surface of the specimen near the notch, is 24.519 nm . 

3-on page 4, lines 131-134, the microstructure, initial grain size, and tensile properties of raw material are presented. So, please refer to an appropriate reference or describe about the obtaining ways of them. 

Answer: The material properties are taken from data bulletin of material producer. Detailed information is enclosed a supplemental file.  The microstructure of the material was studied by EBSD analysis before testing.

Related paragraph changed as following:

Due to the low carbon content, the microstructure consists of 100% ferrite. An EBSD analysis done before the experiment shows an average grain size of 30 µm without particular orientation. The material is tested in its receiving condition. The modulus of elasticity, ultimate tensile stress, and yield stress in order are 207 GPa, 290 MPa, and 240 MPa as provided in supplemental file.

Reviewer 2 Report

The paper deals with a combined crack growth assessment system using thermographic analysis in the apex area together with micrographic findings.
The evaluation is completed by a metallographic analysis by SEM.

I believe it is necessary to use a nomenclature, given that numerous formulas are used, even if they are commonly used in science, considering that these are not always expressed in this form. 

The bibliography, although quite extensive, does not highlight how the correlation between thermography and fatigue since before the 2000s has highlighted the different areas of temperature growth and also in terms of damage and the Wohler curve.

The results express only the values relative to a specimen, stating that other similar tests have been carried out with similar results, but it is not easy to extrapolate the conclusions on the basis of a single load level and relative to a very limited area of the fatigue life. It would be the case that the authors showed the values obtained also from the other tests or did further tests.

The correlation between temperature variations (and the position in which it is measured) and the maximum intrinsec dissipation is not well understood (also in Fig. 6). Comparison with other literature results is similarly difficult due to the different frequencies and different materials.

How and at what stage of the fracture were the fracture images obtained at the SEM?

The Paris curve (Fig. 11) shows two distinct crack initiation values according to the two techniques and two slope values. The explanation should better clarify this difference.

Author Response

1-I believe it is necessary to use a nomenclature, given that numerous formulas are used, even if they are commonly used in science, considering that these are not always expressed in this form. 

Answer: Authors appreciate this suggestion. A nomenclature after the introduction has been added to the publication.

2-The bibliography, although quite extensive, does not highlight how the correlation between thermography and fatigue since before the 2000s has highlighted the different areas of temperature growth and also in terms of damage and the Wohler curve.

Answer: Thanks to point out the issue. Some new references are added to follow up the correlation study between thermography and fatigue. Part of the introduction has been modified accordingly.

When a cracked specimen experiences cyclic loading mode, a cyclic plastic zone appears in front of the crack tip [21, 27]. The deformations produced in the plastic zone in front of the crack yield a temperature rising [28] that can be more easily identified by a thermal camera. Thermographic methods allow to characterize the behaviour of materials [29] subjected to different loads (for example cyclic loads). Thermal effects in fatigue can estimate the fatigue life of materials. The development of the energetic method facilitates the use of infrared thermography in predicting the fatigue behaviour of materials under uniaxial and multiaxial loading with constant amplitude [31-33]. The studies carried out by Bär et al. [34] demonstrated the relationship between the plastic zone and the energy dissipated at the crack tip during fatigue tests.

3-The results express only the values relative to a specimen, stating that other similar tests have been carried out with similar results, but it is not easy to extrapolate the conclusions on the basis of a single load level and relative to a very limited area of the fatigue life. It would be the case that the authors showed the values obtained also from the other tests or did further tests.

Answer:  Thanks for this suggestion. The Authors believe that the current work engaged a complex investigation which involve already many scopes such as crack mechanism, crack path, crack propagation, dissipation, etc. We would like the readership keeping focus on the intrinsic dissipation and crack mechanism, so that the manuscript did not involve the impact from different testing conditions. The similar results with different fatigue stress and fatigue life were provided in the supplemental file.

It is fair that the conclusion is just relative to given condition. The sentence at conclusion section was revised accordingly.

4-The correlation between temperature variations (and the position in which it is measured) and the maximum intrinsic dissipation is not well understood (also in Fig. 6). Comparison with other literature results is similarly difficult due to the different frequencies and different materials.

Answer: Authors thank reviewer for rising the problem here. Paragraph was modified to explain the variation of temperature and intrinsic dissipation in fig 6.

According to the thermodynamic relation in heat equation given above. The temperature evolution at a location relates not only deformation but also temperature field, convection, conduction and radiation. In contrary, the intrinsic dissipation will remain steady if irreversible deformation keeps in the same level. Fig. 6 plots the evolution of maximum value on temperature (in black) and d1  (in blue) as a function of the number of cycles from 7.180×107 to 7.205×107 cycles.  It is found that the average of temperature would slightly increase, while average of d1 remains relatively stable.

Indeed, from the thermodynamic point of view, it is in good agreement that the temperature tends to continuously increasing for the case of unvarying d1 and the negative sign of the Laplacian (see Eq. (1)). Then, both curves (temperature and intrinsic dissipation) sharply increase from 7.205×107 cycles.

When crack comes to the steady growth, the energy dissipation relates mainly on the plasticity in front of crack tip. Since direct measurement on intrinsic dissipation is difficult than calculated from temperature field., authors believe a comparison for the order of intrinsic dissipation with other similar material allows a better understanding of the results.

5-How and at what stage of the fracture were the fracture images obtained at the SEM?

Answer: All SEM images presented in this publication were made after final breakage of the sample. This note has been added in the publication to clarify this.

6-The Paris curve (Fig. 11) shows two distinct crack initiation values according to the two techniques and two slope values. The explanation should better clarify this difference.

Answer: The difference could result from following reason:

  1. The object surface are different side of sample, there for the result would different.
  2. The microstructure significantly influence the fatigue initiation and early propagation rather than long crack propagation, so that difference in microstructure on the two observed side surface would cause the differences.
  3. As from fractography, the fronter of a crack at transition from initiation to propagation has a curved shape rather than a straight line. Simplification on geometry calculation could cause the different
  4. two techniques associated with its processing method could results also a slight difference

The corresponding paragraph has been revised to clarify the results obtained and explain the difference.

Fig. 11 shows the fitting of the experimental data of the FCGR in terms of the stress intensity factor based on the optical camera measurements and the d1 approach in a double logarithm coordinate system. The dotted lines correspond to the transition from the fractographic images in Fig. 7(a). According to Fig. 7(a), the transitions (initiation to short cracking) appear at 252 microns for the optical camera and 181 for the infrared camera. Furthermore, based on Eqs. (2) and (3), the ∆K of the transition lengths are at 3.92 MPa.m1/2 for the d1 results and 3.29 MPa.m1/2 for the optical camera results. Whereas the fatigue tests performed by Pippan et al. [61] on Armco iron revealed the existence of a threshold value of ∆K at 2.75 MPa.m1/2.  The transition between the short and the long cracking (at 1409 µm) is equivalent to a ∆K of 15.0 MPa.m1/2. These results allow us to establish the dotted lines, as presented in Fig. 11. A trend line calculation was performed on the part associated with the short cracking. The results, expressed in the power form, are characterized by the well-known Paris equation, da⁄dN=C.∆Km , where the parameters C and m are the material constants [60]. The results for each method are obtained as follows:

  • From d1: da⁄dN=1E-11∆K 24
  • From the optical camera: da⁄dN=2E-11∆K 01

The value of the exponent m in Fig.11 is 3.24 (in blue) for the optical method and 3.01 for the d1-based methodology (in red). Besides, the values indicated in Fig 11 are from two side surfaces of sample by optical or Infrared camera respectively.

The Authors would like to thank the Reviewer for their valuable comments and hope that the changes matched the comments. If Reviewer believe that additional changes are necessary, I will be glad to make them.

Round 2

Reviewer 2 Report

In my opinion, the request for revision envisaged the development of further tests to validate the conclusions, therefore certainly longer times. However, the review has been carried out and the responses to the remarks are correct. Considering the material of the additional file, therefore, I consider that this work can be published, hoping that in the future the authors will continue the research leading to statistically more reliable results.